# Zero-Shot Simultaneous Multislice Reconstruction Based on Pretrained Generative Image Priors

**Kexin Yang**[1]                                                            2100411015@stumail.sztu.edu.cn
**Shoujin Huang**[1]                                                       solor.pikachu@gmail.com
**Guanxiong Luo**[2]                                      guanxiong.luo@med.uni-goettingen.de
**Lifeng Mei**[1]                                                              meilifeng@sztu.edu.cn
**Jingyu Li**[*1]                                                              lijingyu@sztu.edu.cn

[1] *Shenzhen Technology University, Shenzhen, China 1*

[2] *University Medical Center Göttingen, Göttingen, Germany 2*

**Editors:** Accepted for publication at MIDL 2024

## Abstract

In this study, we combine the readout-concatenation framework with generative image priors to achieve simultaneous multislice imaging (SMS) reconstruction. The results show that generative image priors have better generalization than supervised deep learning methods such as VarNet, and it can be processed flexibly handle arbitrary slice aliasing patterns with in-plane acceleration.

**Keywords:** Simultaneous Multislice Imaging, Reconstruction, Deep Learning, Generative Image Priors

## 1. Introduction

Simultaneous multislice imaging (SMS) (Barth et al., 2016) has undergone significant evolution, emerging as a technology for accelerating Magnetic Resonance Imaging (MRI). In addition to conventional reconstruction techniques like SENSE (Pruessmann et al., 1999), GRAPPA (Griswold et al., 2002), and Compressed Sensing (CS) (Lustig et al., 2007), deep learning-based methodologies have been noted for SMS image reconstruction. Traditional reconstruction methods exhibit a vulnerability to noise amplification at high acceleration factors, whereas certain supervised deep learning approaches lack the requisite generalizability, rendering them less effective when confronted with diverse datasets, such as VarNet (Sriram et al., 2020). In recent times, techniques based on image priors (Luo et al., 2023) have demonstrated commendable robustness; however, their application in the context of SMS reconstruction remains underutilized in the academic community. To fill this gap, we combine the readout-concatenation framework (Moeller et al., 2010) with generative image priors method to achieve SMS reconstruction. This method can flexibly deal with arbitrary in-plane accelerated slice aliasing mode while maintaining good stability.

---

* Corresponding author

## 2. Methods

**SMS-SPRECO reconstruction.** The SMS reconstruction method we used is based on the speed up MR scanner with generative priors for image reconstruction (SPRECO) (Luo et al., 2020) and readout-concatenation framework.

As shown in Figure 1, the aliased raw SMS data are concatenated in the readout direction and subsequently processed to generate data for the extended field of view (FOV). At the same time, the coil sensitivity map (CSM) is estimated by BART (Blumenthal et al., 2023) from the pre-scan calibration data. Then, the CSM is employed to perform coil expansion and coil reduction operations. The expansion operation extends the image to multi-coil representation, while the data consistency (DC) layer integrates portions of the originally readout-concatenated k-space into the reconstructed k-space. Following this, the reduction operation combines the image coils. We utilize a diffusion-denoised probabilistic model (DDPM) as the prior, and Gaussian noise initialized undergoes T-step reverse diffusion process with guidance from the DC.

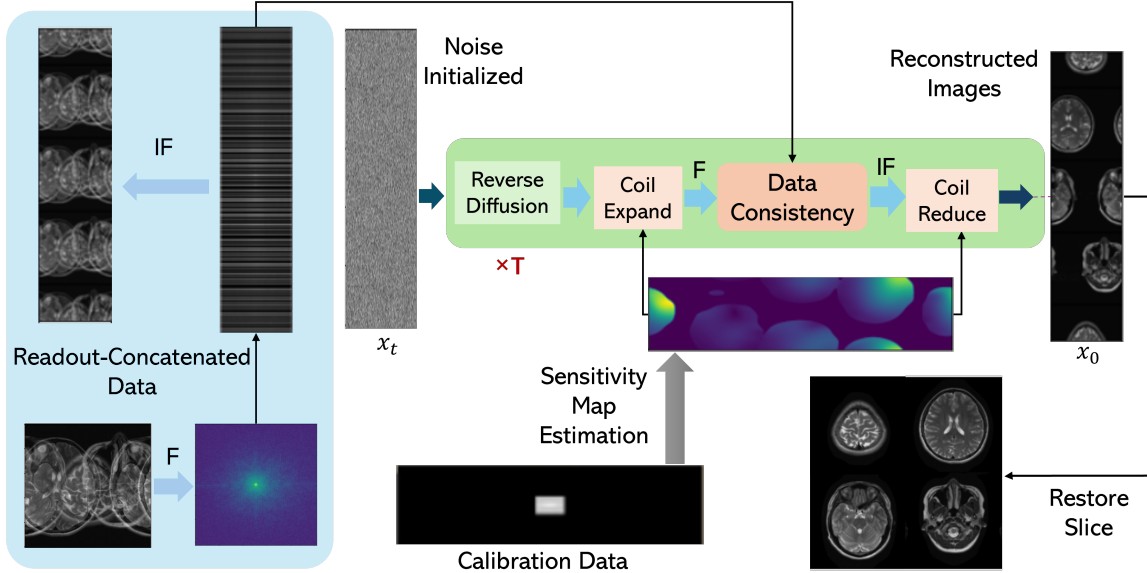

Figure 1: The schematic illustration of SMS-SPRECO reconstruction approach. Diffusion model is applied and the data consistency term of readout-concatenated k-space is integrated as the guidance for the generative reverse process.
F: Fourier transform; IF: Inverse fourier transform.

**Experiments on acquired data.** We gathered a dataset comprising 17 EPI acquisitions at The First Affiliated Hospital of Tsinghua University in Beijing of China. The data was acquired using GE's SIGNA Premier 3.0T magnetic resonance imaging system. Among the specified acquisition parameters, the matrix size was configured at 128x128, with 48 imaging slices, and the number of receiver channels was automatically determined by the machine, typically comprising 18 channels. Moreover, there exist additional data preprocessing operations for EPI data, including ghost correction, rampling sampling correction.

**Details of the experiments.** In our experiment, we employed various methods of comparison, including the supervised deep-learning method E2E-VarNet, as well as traditional reconstruction methods such as GRAPPA and L1-PICS.

The SMS-SPRECO we used is one of the generative image priors pretrained by external T1, T2 and FLAIR data, while E2E-VarNet methods were trained with mixed MB factors ranging from 2 to 5 randomly in combination with in-plane acceleration of 1, 2, and 4 at fastmri dataset emulated SMS data, and then fine-tuned using the EPI data.

## 3. Results

As the reconstructed EPI-SMS images under multiband at 2 and in-plane acceleration factors at 2 showing in Figure 2, the SMS-SPRECO method exhibits strong generalization and closely aligns with GRAPPA and L1-PICS. By contrast, Varnet, a supervised deep learning approach trained on similar data, performs less effectively than our method, which achieves results comparable to those of the best traditional reconstruction methods under zero-shot conditions. This demonstrates the strong generalization capability of our approach.

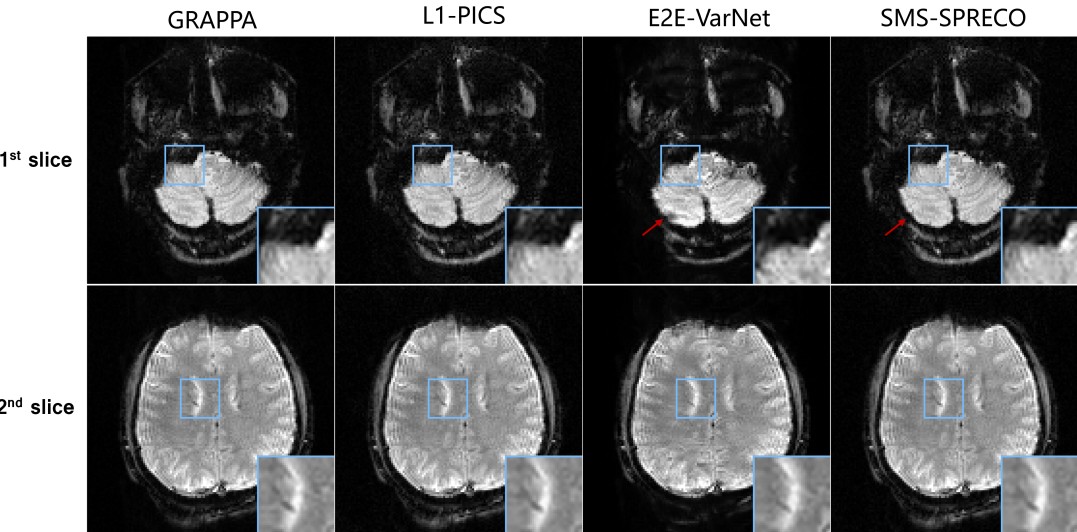

Figure 2: Visual comparison of EPI-SMS reconstruction results for Multiband Factor at 2 and acceleration factors at 2.

## 4. Discussion and Conclusion

In summary, we apply SPRECO to SMS reconstruction, and the results demonstrate that the method is robust and applicable to SMS data. Furthermore, with improvements in coil sensitivity estimation, this approach has the potential to achieve even better results, highlighting its significant promise.

## Acknowledgments

This study is supported in part by Natural Science Foundation of Top Talent of Shenzhen Technology University (No.GDRC202134) and the National Natural Science Foundation of China (No.62101348).

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
