# OpenReview forum: "Zero-Shot Simultaneous Multislice Reconstruction Based on Pretrained Generative Image Priors"
_MIDL.io/2024/Short_Papers — MIDL 2024 Short Papers_

### Official Review · Reviewer_jomK · 2024-04-24

**Confidence:** 3
**Final Rating:** 3.5

**Review:**

The authors combine an Simultaneous Multi-Slice (SMS) MRI image formation model with a deep prior (a diffusion model). This is, to the best of my knowledge, the first paper to achieve this with SMS acquisitions, which are widespread in diffusion and functional MRI. The method is well justified (if anything, I miss references to the work from Jon Tamir's group) and the paper is well written.

The main drawback is that the results are just qualitative and the differences between the methods seem tiny.

---

### Decision · Program_Chairs · 2024-04-26

Accept